# Changes in Body Weight and Risk Factors for Overweight and Obesity in 5–6-Year-Old Children Attending School in Geneva

**DOI:** 10.3390/children11050529

**Published:** 2024-04-28

**Authors:** Luisa Narvaez, Per Bo Mahler, Denise Baratti-Mayer, Emilien Jeannot

**Affiliations:** 1Institute of Global Health, Faculty of Medicine, University of Geneva, Chemin de Mines 9, 1202 Geneva, Switzerland; lufenarvaez@hotmail.com; 2School Health Service, Department of Public Instruction, 1207 Geneva, Switzerland; per.mahler@gmail.com (P.B.M.);; 3Addiction Medicine, Department of Psychiatry, Lausanne University Hospital, 1011 Lausanne, Switzerland

**Keywords:** childhood, prevalence, obesity, overweight, body mass index, COVID-19

## Abstract

**Background:** The prevalence of obesity and overweight in children is increasing in industrialized countries. Monitoring the evolution of these phenomena is essential for understanding prevention and health promotion programs. This study aims to present the analysis of anthropometric data collected by school nurses from the School Health Service of Geneva (Service de santé de l’enfance et de la jeunesse) for children aged 5 to 6 years during the 2021–2022 school year, as well as describe the trends in overweight and obesity from 2003–2004 to 2021–2022. Risk factors were also assessed in the 2021–2022 sample. **Methods:** This study included a random sample of 958 (479 girls and 479 boys) primary school pupils aged 5 to 6 years in Geneva. Data on weight, height and socioeconomic status were collected. BMI was analyzed using the Cole standard. A multivariate analysis was conducted to assess the influence of socioeconomic factors on overweight and obesity. We compared these results with BMI trends in students of the same age since 2003. **Results:** In 2021–2022, overall prevalence of overweight was 12.73%, and obesity was 5.64%. Girls had higher rates of overweight (14.20%) and obesity (6.68%) compared to boys (11.27% and 4.59%, respectively) (*p* < 0.0001). Overweight in boys significantly increased since the 2013–2014 and 2019–2020 measurements (*p* = 0.003). The trend for girls was similar but not statistically significant. Obesity rates have not significantly increased since 2019–2020 in both genders, but there is a significantly increasing trend for girls since 2013–2014 *p* = 0.045). Socioeconomic factors, particularly the socioeconomic class of parents, played a predictive role in overweight and obesity. **Conclusions:** The School Health Service of Geneva and the Directorate General of Health have a crucial role in monitoring and preventing childhood obesity. The prevalence of overweight and obesity has remained high since 2010, justifying continuous efforts for prevention. A significant increase in prevalence has been observed since 2020, particularly among overweight boys, and could be related to COVID-19 confinement measures.

## 1. Introduction

The prevalence of obesity and overweight in children is increasing in industrialized countries [1,2,3]. In 2022 the Lancet Child and Adolescent Health published a systematic review of global data on obesity and overweight that shows that the prevalence of overweight and obesity has increased significantly in children and adolescents in developed countries, accompanied by a sharp increase in the metabolic syndrome. The study estimates that the global prevalence of the metabolic syndrome in 2020 is at 2.8% in children and 4.8% in adolescents (25 million children and 35 million adolescents). For children, the prevalence of the metabolic syndrome was found to be at 2% in high-income countries and at 3% in upper-middle-income countries [4]. Other studies have also shown a high prevalence of overweight and obesity. Among children aged 5–17 years, the prevalence of overweight and obesity is equal to or greater than 30% in Greece (44% in boys and 38% in girls), Italy (36% in boys and 34% in girls), and New Zealand (34% in boys and 34% in girls) [5].

In 2016, the World Health Organization (WHO) estimated that 41 million children under 5 years of age were overweight or obese. The prevalence of overweight and obesity among children and adolescents aged 5–19 has increased significantly between 1975 and 2016, going from 4% in 1975 to just over 18% in 2016 [6].

In Europe, the prevalence of overweight and obesity in children and adolescents has been increasing since the 1980s. This trend has, however, stabilized or slightly decreased in Northern and Central Europe between 1999 and 2016, but has continued to increase in Southern Europe [7,8].

Scientific evidence shows that the consequences of being overweight in childhood and adolescence can be numerous and severe, including endocrine, cardiovascular, renal, pulmonary, orthopedic, and gastroenterological diseases [9,10,11]. Low levels of self-esteem due to stigmatization have also been noted, which can lead to reduced socialization and learning difficulties [12,13,14].

Assessing the global obesity epidemic is made difficult by the use of different reference systems to define overweight and obesity in children, resulting in different estimates of prevalence. This problem is compounded by the use of different anthropometric methodologies in the absence of agreement on a standardized protocol. Childhood obesity is generally defined on the basis of body mass index (BMI = Weight/Height^(2)^). There are different references: national curves and American curves from the Center for Disease Control (CDC). They enable clinicians to monitor changes in children’s weight. They can also be used to define obesity. The IOTF (International Obesity Task Force) thresholds are recommended for establishing protocols based on a common definition. There are other methods for assessing obesity in children and adolescents. Fat mass can be measured. The main techniques for measuring this indicator involve measuring skinfolds at different sites and calculating brachial surface areas from skinfolds and arm circumference. The ratios of skin folds or circumferences (waist, hip, thigh) predict the distribution of body fat.

Because of the serious public health consequences of being overweight and obese, it is of paramount importance to monitor trends among the youth. This permits us to evaluate the effect of public health interventions in schools, based on healthy eating and exercise promotion, and helps us to make the interventions as effective as possible in a constantly changing environment.

In this study, another variable, which led to widespread modifications in behavior, was the COVID-19 pandemic. This real-life situation permitted us to see the influence of reduced attendance to school, global reduction in physical activity, and to some extent changes in eating habits, on BMI. Considering our data have shown a global stabilization of overweight and obesity in youth for the last 10 years, these data are of utmost interest.

This study aims to present the analysis of anthropometric data collected by school nurses from the School Health Service of Geneva for children aged 5 to 6 years during the 2021–2022 school year, as well as observe the trends in overweight and obesity between 2003 and 2022. Some risk factors for overweight and obesity (SES and nationality) were also assessed in the 2021–2022 sample.

## 2. Materials and Methods

### 2.1. Subjects

Beginning in 2003–2004, the School Health Service of Geneva (Service de santé de l’enfance et de la jeunesse) has been monitoring children’s body mass index (BMI) of 5–6-year-old children attending public schools. These measures were originally part of a standard health evaluation of children starting school, which was set up by the department of education in the early 20th century. The monitoring, started in 2003, was aimed to determine the prevalence of overweight and obesity in youth and their evolution over time and identify potential risk factors [15,16].

Data were collected from ten methodologically similar cross-sectional surveys of schoolchildren aged 5–6 years, enrolled in the public school system starting in 2003–2004. Representative samples were used for the 2017–2018 and 2021–2022 school years. The sample consisted of children from primary schools, randomly selected by the Service for research in education of the canton of Geneva (SRED). For the 2017–2018 school year, it was impossible to collect data on the totality of the age group for logistical reasons and because of a change in the organization of health visits. For the 2021–2022 school year, in addition to assessing weight and height, socioeconomic variables were also included to look for potential risk factors associated with being overweight or obese.

A total of 27,281 schoolchildren (13,874 boys and 13,387 girls) aged 5–6 years were screened during the 2003–2022 period. Children attending private schools in Geneva were not included in the study. A full description of the methodology has been published previously [15,16].

### 2.2. Socioeconomic Variables

The socioeconomic variables included were the passport nationality and the socio-professional category of the mother and the father. These data were provided by the SRED (https://www.ge.ch/organisation/service-recherche-education, accessed on 10 March 2020) using an official government database.

Nationality corresponded to the country that was indicated on the child’s passport and was classified into the following groups: Switzerland, Portugal, Italy, France, and Kosovo (N > 30). The other categories were other European countries, America (North, Central, and South), Africa, and other (Oceania and Middle East).

The parental socioeconomic status was divided into the five categories used by the SRED: leaders and senior executives, executives and employees, self-employed, workers, and “other”. The unemployed, or those whose occupations did not correspond to one of the previous four categories, were included in the “other” category. Parents whose occupation remained unknown were also included in the “other” category.

### 2.3. Anthropometric Measurements

The methodology and equipment were identical across the ten surveys. School nurses recorded the date, date of birth, gender, weight, and height of all children using standard anthropometric techniques [17].

The children were measured in gym clothes and without shoes. Body weight (kg) was measured using a calibrated SECA™ digital scale (SECA™ Alpha, Model 770, Hamburg, Germany). Height (cm) was measured during inspiration using a wall-mounted or balance-mounted stadiometer.

The collected data were recorded on an ad hoc platform to calculate each child’s body mass index (BMI), and then assigned to one of three categories: normal, overweight, or obese. IOTF thresholds were used to determine the BMI cutoff points for determining in what category (normal, overweight, and obese) the children fall in. The International Obesity Task Force has developed a definition of overweight and obesity in children, using BMI curves based on data collected in six countries with large representative samples. Available from the age of 2 to 18, the thresholds for overweight and obesity are the percentile curves reaching the values of 25 and 30 kg/m^2^, respectively, at the age of 18. According to the IOTF definition, overweight (including obesity) is defined as a BMI above the IOTF-25 centile, overweight (not obese) as between the IOTF-25 and IOTF-30 centiles, and obesity as a BMI above the IOTF-30 centile. We have been using this method since the beginning of our data collection and we estimate it best reflects the profile of our population and the recommendations in the literature [18].

### 2.4. Statistical Analysis

The data for both boys and girls were grouped for analysis and then analyzed separately. All categorical variables were expressed in terms of numbers of cases and percentages. The Kolmogorov–Smirnov test provided the normal distribution of the variables (height and weight).

Comparisons between the prevalence of overweight and obesity and the evolution of the prevalence rates were made through the chi-square and Fisher exact tests. Time trends were assessed using a linear regression. To identify risk factors, a multivariate logistic regression model was used to investigate the relationship between obesity or overweight and potential socioeconomic risk factors. The final model was selected using a stepwise procedure according to the Akaikes method, and the results were expressed as odd ratios (OR) with confidence intervals (CI). A significant difference threshold of 0.05 was considered.

### 2.5. Ethics

When a child starts school in a public school in Geneva, parents are informed in writing that their children will periodically be weighed and measured to monitor BMI trends in the canton. Parents can therefore refuse the measurements and not have their children measured. It is noteworthy that very few parents (<1%) refused the measurement. The monitoring of BMI in Geneva and this study have been accepted by the Department of Public Education of the canton of Geneva (DIP-REQ 2020).

## 3. Results

Table 1 shows the prevalence of normal, overweight, and obesity in children aged 5–6 years in the canton of Geneva for both genders and for the years between 2003–2004 and 2021–2022. The overall prevalence of overweight stayed stable at 10.3% between the 2003 and 2004 school year and until 2017–2018 [16]. An increase of 2.43 points (*p* = 0.003) was then observed between 2017–2018 and 2021–2022 (Figure 1). Regarding obesity, the prevalence has increased by 2.04 points (*p* = 0.002) since 2003–2004, with a large increase (3.24 points) since 2010–2011.

### 3.1. Results for the 2021–2022 School Year and Important Changes since 2010–2011

The data collected during the 2021–2022 school year, after the COVID-19 pandemic, on a sample group of 958 children, (479 girls and 479 boys) is shown in Table 1. The overall prevalence of overweight was 12.73% and 5.64% for obesity. By gender, it was 14.20% and 6.68% for girls and 11.27% and 4.59% for boys (Table 1). The intergender difference was statistically significant (*p* < 0.0001) for overweight but not for obesity.

Our data show a highly significant increase in overweight among boys from 5.90% in 2017–2018 to 11.27% in 2021–2022 (*p* < 0.001). Since the beginning of BMI data collection in Geneva (2003), there has been an increase in overweight among boys (Ptrend = 0.036); with a period of significant decrease from 2008–2009 to 2017–2018 (3.5 points, *p* = −0.0287) [16]. For girls, the trend is less marked than for boys, even showing a slight non-significant decrease in the prevalence since 2017–2018 going from 14.50% to 14.20% (Figure 1).

Regarding obesity, an increase was mainly seen among girls, with a 2.2 points increase from 4.5% to 6.7% between 2017–2018 and 2021–2022 (*p* = 0.12). The evolution since 2003–2004 also shows a statistically significant progressive increase, (Ptrend = 0.006) with a progressive increase since 2010–2011, the year when the lowest obesity prevalence was measured. For boys, there is a slight, but not significant, increase in obesity between 2017–2018 and 2021–2022 of 0.3 points (*p* = 0.8). If we consider the trend since the beginning of our data collection, we observe a non-significant increase in the prevalence of obesity among boys (Ptrend = 0.25) (see Figure 2).

### 3.2. Risk Factors for Being Overweight or Obese in the 2021–2022 School Year

Table 2 shows the socioeconomic factors potentially related to being overweight or obese for the 2021–2022 sample. The girls are 1.4 times more likely to be overweight or obese than the boys (AOR = 1.5 95% CI 1.1–1.9). Regarding overall overweight and obesity, the risk factors related to the parents’ socio-professional category show that the children whose mothers belong to the “other” category (unemployed or not belonging to the other categories) are 4.8 times more likely to be overweight or obese than the children whose mothers are managers (AOR = 4.8 CI 2.05–11.56). The children whose fathers are self-employed (AOR = 2.5 CI 1.3–6), workers (AOR = 2.03 CI 1.9–3.04), or in the “other” category (AOR = 2.7 CI 1.36–4.2) are statistically more likely to be overweight or obese than the children whose fathers are in the “manager” category. No statistically significant association was found for nationality.

## 4. Discussion

This present study highlights the trends in the prevalence of overweight and obesity in children aged 5 to 6 years in the canton of Geneva starting in 2003–2004 and up to 2021–2022. The analysis focuses on the follow-up of our previous publication for the 2017–2018 school year [16]. The latest data were collected in the immediate post-pandemic period and included socioeconomic variables potentially related to being overweight or obese.

The data leading up to the 2017–2018 school year have shown considerable stability in the prevalence of being overweight since 2003–2004. A small, but statistically significant, increase in overweight was observed in the boys and was less marked for the girls. A slight, non-significant, increase in obesity was observed in the boys and was slightly more marked and statistically significant in the girls. However, since the 2010–2011 school year, when the lowest prevalence of obesity was reported, there has been a significant increase.

The data from 2021–2022, immediately following the COVID-19 pandemic, show a relatively marked increase in overweight and obesity in both the boys and girls. Regarding overweight, there was a statistically significant increase in the boys and a slight non-significant decrease in the girls. For obesity, there was a non-significant increase in both sexes, which was more marked in the girls than in the boys.

A number of hypotheses can be proposed to explain these changes. For the prevalence of overweight, the pandemic period seems to have had an important influence on BMI in both sexes. The increase is considerably more marked in the boys, who maybe suffered more from inactivity than the girls and started at a lower relative average BMI. For the girls, on the other hand, considering the important increase in obesity, it is possible that the relative decrease in overweight measured is due to the girls having moved up from overweight to obesity which might not have been compensated by a sufficient BMI increase in the rest of the group.

Discussions with parents, following the pandemic, seem to point to changes in eating habits; the children being at home tend to snack a lot more and compensate for boredom with food, as well as changes in physical activity (no school gym or no club sports) even though Switzerland was relatively “lenient” to potential outdoor activities during the COVID.

In comparison to international overweight and obesity prevalence rates, we can observe that the situation in Geneva, although preoccupying, is considerably less worrying than in some other industrialized countries, even considering the increase in recent years.

For example, the data from the Centers for Disease Control in the USA indicated that for children and adolescents aged 2 to 19 years, the prevalence of obesity in 2017–2018 was 19.3%. The prevalence of obesity was 13.4% in children aged 2–5 years, 20.3% in children aged 6–11 years, and 21.2% in children aged 12–19 years [18]. It was also observed that the prevalence of obesity was 25.6% in Hispanic children, 24.2% in non-Hispanic black children, 16.1% in non-Hispanic white children, and 8.7% in non-Hispanic Asian children.

The latest report from WHO Europe shows that obesity and overweight rates vary widely across Europe. The report presents data collected between 2015–2017 (pre-COVID-19) on children aged 6 to 9 years in 36 member states of the European region (about 250,000 children). The prevalence of overweight (including obesity) was 29% in boys and 27% in girls, and obesity was 13% and 9%, respectively. Mediterranean countries such as Cyprus, Spain, Greece, and Italy have the highest proportions of childhood overweight and obesity in Europe: more than 40% of boys and girls are overweight, and 19 to 24% of boys and 14 to 19% of girls are obese. Northern countries have lesser rates, although Estonia, Finland, France, Kyrgyzstan, Latvia, Slovenia, and Sweden have an increasing trend in both genders. There has, however, been a decreasing trend in 5 (Greece, Italy, Portugal, Slovenia, and Spain) of the 13 countries monitored since 2007. The reduction in the prevalence of overweight varies from 4 to 12 percentage points for boys and 3 to 7 for girls. The prevalence of obesity shows a similar trend [19].

Obesity and overweight among children is not just a problem in rich countries, as this study explains. For several years now, obesity rates have been rising faster in low- and middle-income countries than in high-income countries, in both urban and rural areas. These are the findings of a study by the International Fund for Agricultural Development (IFAD). Through a study of five countries—Egypt, Indonesia, Nigeria, Bolivia, and Zambia—IFAD is alerting us to the fact that this is a major public health issue throughout the world. Defined by the World Health Organisation (WHO) as a body mass index (BMI) greater than or equal to 30, obesity in these countries has risen from around 15% in 1992 to 44% in 2015 in rural areas, while obesity rates in the urban population have risen from 36% to 51%. The authors observe a gradual increase in obesity in the five countries studied, even though daily calorie intake per person has stagnated since 2010. This suggests that other factors are at play, such as changing lifestyles and eating habits, explains the authors. In low-income countries such as Egypt and Bolivia, economic growth is associated with increases in BMI, while in high-income countries, economic growth is correlated with a lower BMI [20]. We can observe the same trend in Brazil. In this country, the prevalence of overweight/obesity is from 26.8% to 30% in boys and 23.9–26.6% in girls aged between 5 and <10 years [21].

Increases in overweight and obesity during/following the pandemic have also been observed in other studies. A large cohort study of 432,302 people aged 2–19 years in the USA showed a significant increase in overweight and obesity between 2018 and 2021 [22]. In France, a study of 50,000 children enrolled in kindergarten showed that the proportion of obese children almost doubled during the two years of the pandemic, rising from 2.8% to 4.6%. The rate of overweight children also increased from 8.9% to 11.2% [23].

For the whole of Switzerland, the latest BMI monitoring report published by Health Promotion Switzerland (2019–2020) found a relatively little “COVID effect” in other cantons. It will be interesting to see if the future inter-cantonal comparisons show any “delayed” COVID-19 effect [24]. Historically, Geneva has shown similar trends to the rest of the country and it was a surprise to see this difference, which we fail to explain for the time being.

Risk factors that were statistically correlated to overweight and obesity were the female gender and the SES of the parents. These factors have already been described in the literature as risk factors for overweight and obesity and were mentioned in our previous study [25]. We did not, however, find any correlation with nationality, which was observed in a study done in the canton of Zurich. This could be due to the small sample size or might underline the relatively more important effect of SES compared to nationality [8].

### Limits of the Study

The limits of this study might include the sampling of the 2017–2018 and 2021–2022 subjects, which could affect the data, giving a partial picture of the population. This study only concerns children attending public schools in the canton of Geneva and does not include children attending the private schools (15% of the population), who tend to be in a higher SES group. Using BMI to evaluate the fat mass instead of a more precise measurement (skinfold) and other anthropometric measurements that would have enabled a more complete and relevant assessment of overweight and obesity in our population is also a limitation [26]. Limiting the socioeconomic variables to nationality and the parents’ professional status, while focusing on other variables such as being a mono-parental family, both parents having a professional activity, the child’s regular sports activities, and daily time on screens, amongst others, could permit us to better understand this global health problem. Finally, we are confronted with the limitations of a cross-sectional study. A cohort study is more appropriate but also more complicated to set up.

## 5. Conclusions

The BMI data collected among 5–6-year-old children, in the canton of Geneva, during 2021–2022 show that there have been substantial changes in the percentage of overweight and obese children over the last 20 years, in particular during the last 3 years which include the pandemic period. This change in dynamics is of concern, given the relative stability of this prevalence over the last 10 years before 2018.

A clear difference was observed between the boys and girls showing an acute increase in obesity among the girls and in overweight among the boys. This difference might be explained by the higher prevalence of overweight in the girls, compared to the boys, before the pandemic. The “at risk” girls have moved from the overweight to the obese group. The overall increase in BMI during this period might be explained by the COVID-19 pandemic, considering the relative stability in the prevalence of BMI before the pandemic. A “COVID effect” has also been observed in other studies and in other countries.

According to parents, the children who were kept home from school ate differently (more snacking and compensation for lacking socialization) and participated less in physical activity and sports due to the closure of schools and sports clubs.

Female gender and parental SES were shown to correlate with excessive weight gain, which has already been described in the literature. Nationality did not correlate with excessive weight in our study.

It will be interesting to see the future evolution in the prevalence of overweight and obesity to see if there can be a return to pre-COVID-19 levels, hoping for a return to a healthier active lifestyle.

These findings might underline what a crucial role schools have in maintaining a healthy lifestyle and will help us to adapt our school-based preventive programs in the hope of minimizing the effect of problems related to excessive weight gain.

## Figures and Tables

**Figure 1 children-11-00529-f001:**
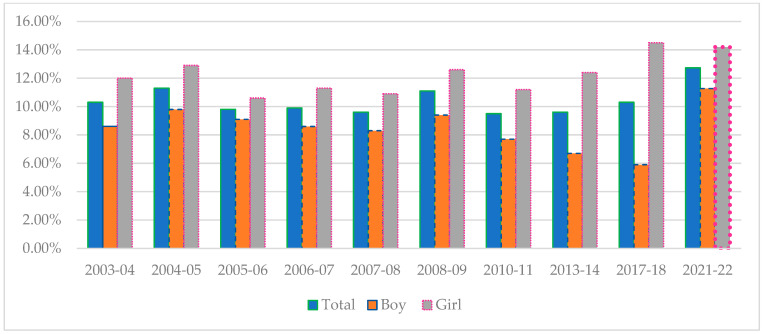
Changes in overweight by gender between 2003–2004 and 2021–2022.

**Figure 2 children-11-00529-f002:**
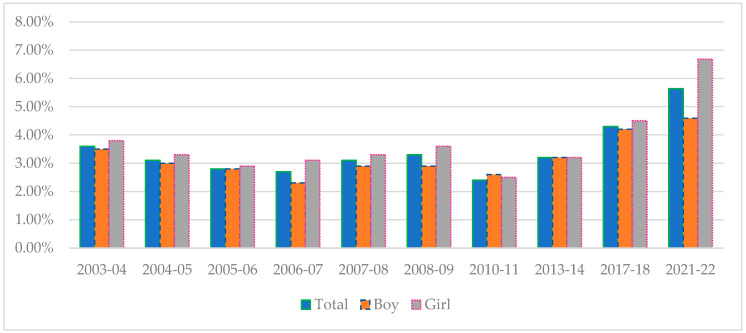
Changes in obesity by gender between 2003–2004 and 2021–2022.

**Table 1 children-11-00529-t001:** Summary of normal weight, overweight, and obesity prevalence by gender between the school years 2003–2004 and 2021–2022.

	Entire Group				Boys				Girls			
Survey Year	*n*	Normal	Overweight	Obesity	*n*	Normal	Overweight	Obesity	*n*	Normal	Overweight	Obesity
		ISO-BMI < 25 kg/m^2^	ISO-BMI 25–30 kg/m^2^	ISO-BMI > 30 kg/m^2^		ISO-BMI < 25 kg/m^2^	ISO-BMI 25–30 kg/m^2^	ISO-BMI > 30 kg/m^2^		ISO-BMI < 25 kg/m^2^	ISO-BMI 25–30 kg/m^2^	ISO-BMI > 30 kg/m^2^
2003–2004	3728	86.10%	10.30%	3.60%	1926	87.90%	8.60%	3.50%	1802	84.20%	12.00%	3.80%
2004–2005	3908	85.60%	11.30%	3.10%	1981	87.20%	9.80%	3.00%	1927	83.80%	12.90%	3.30%
2005–2006	3718	87.40%	9.80%	2.80%	1884	88.10%	9.10%	2.80%	1834	86.50%	10.60%	2.90%
2006–2007	3109	87.40%	9.90%	2.70%	1934	89.10%	8.60%	2.30%	1175	85.60%	11.30%	3.10%
2007–2008	3533	87.30%	9.60%	3.10%	1769	88.80%	8.30%	2.90%	1764	85.80%	10.90%	3.30%
2008–2009	2609	85.60%	11.10%	3.30%	1186	87.70%	9.40%	2.90%	1423	83.80%	12.60%	3.60%
2010–2011	2600	88.10%	9.50%	2.40%	1235	89.70%	7.70%	2.60%	1365	86.30%	11.20%	2.50%
2013–2014	2194	87.20%	9.60%	3.20%	1026	90.10%	6.70%	3.20%	1158	84.40%	12.40%	3.20%
2017–2018	924	85.4	10.30%	4.30%	454	89.90%	5.90%	4.20%	470	81.00%	14.50%	4.50%
2021–2022 ^1^	958	81.63%	12.73%	5.64%	479	84.13%	11.27%	4.59%	479	79.12%	14.20%	6.68%
TOTAL 2003–2022	27,281				13,874				13,397			

^1^ These numbers show a clear difference between the 2017–2018 values and the latest collection.

**Table 2 children-11-00529-t002:** Socioeconomic risk factors for being overweight and/or obese in the 2021–2022 sample.

	OR	Adjusted OR	Confidence Interval 95%
**Gender**			
Boy	1	-	-
Girl	**1.4**	**1.5**	**1.1–1.91**
**Nationality**			
Switzerland	1	-	-
Portugal	1.36	1.38	0.73–2.5
France	1.01	1.12	0.47–2.15
Kosovo	1.23	1.27	0.54–2.76
Italy	0.94	0.91	0.47–1.86
Other European	1.73	1.8	0.78–3.84
America (North, Central, and South)	1.68	1.75	0.72–3.88
Africa	0.51	0.64	0.21–1.22
Other (Middle and Far East, Oceania)			
**Mother’s socio-professional categories**			
Leaders and senior executives	1	-	-
Executives and employees	0.81	0.91	0.56–1.18
Self-employed	1.21	1.23	0.24–6.14
Workers	1.33	1.3	0.80–2.19
Others ^1^	**4.8**	**4.6**	**2.05–11.56**
**Father’s socio-professional categories**			
Leaders and senior executives	1	-	-
Executives and employees	1.13	1.16	0.66–1.8
Self-employed	**2.8**	**2.5**	**1.3–6**
Workers	**2.01**	**2.03**	**1.19–3.04**
Others ^1^	**2.4**	**2.7**	**1.36–4.2**

^1^ Unemployed, unknown occupation, or not belonging to the other categories.

## Data Availability

The data presented in this study are available on request from the corresponding author due to restrictions of privacy.

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
