# Peer review of "Changes in Body Weight and Risk Factors for Overweight and Obesity in 5–6-Year-Old Children Attending School in Geneva"

_children, 2024, doi:10.3390/children11050529_

Round 1
Reviewer 1 Report
Comments and Suggestions for Authors
The article addresses the important issue of overweight and obesity among children.
Below are my comments and suggestions:
• Include at the end of the introduction what new things the research brings to science. Please emphasize the sense and value of carrying them out.
• In my opinion, section 2.4 is not necessary and information on BMI classification could be included in section 2.3. Anthropometric measurements and assessment of nutritional status
• Why was the BMI classification not based on the WHO percentile charts?
• Incorrect numbering of subchapters - chapter 2.4 appears twice
• Has the consent of the relevant Ethics Committee been obtained for the research?
• Table 1 – missing units for BMI
• Figure 1 – if units (%) are provided in the axis title, there is no need to include units on the chart. Additionally, I suggest starting OY with 4%, then the chart will be more readable. Similarly Figure 2.
• Discussion is insufficient. First of all, the impact of socio-economic factors should be described in more detail.
• Delete point 6. Patents
Author Response
Thank you for your comments. We have tried to address the issue you have raised, as far as possible.
Comment 1 : • Include at the end of the introduction what new things the research brings to science. Please emphasize the sense and value of carrying them out.
Answer 1 : We have added more information on the latest scientific developments on this subject in the introduction.
Comment 2 : « In my opinion, section 2.4 is not necessary and information on BMI classification could be included in section 2.3. Anthropometric measurements and assessment of nutritional status”
Answer 2 : We have modified les section 2,4 et 2,3 pour répondre à ce commentaire.
Comment 3 : Why was the BMI classification not based on the WHO percentile charts?
Answer 3 : We have chosen to analyse these data using the reference of the school and not that of the WHO percentile charts in order to maintain continuity in the method of analysis with the subsequent studies to which we are comparing ourselves.
Comment 4 : Incorrect numbering of subchapters - chapter 2.4 appears twice
Answer 4: We have modified this error
Comment 5 : Has the consent of the relevant Ethics Committee been obtained for the research?
Answer 5: Yes, we have written a section (2.5) detailing how this study was accepted by the Geneva ethics committee.
Comment 6 : Table 1 – missing units for BMI
Answer 6: We have added the unit for BMI
Comment 7 : Figure 1 – if units (%) are provided in the axis title, there is no need to include units on the chart. Additionally, I suggest starting OY with 4%, then the chart will be more readable. Similarly Figure 2.
Answer 7: We have modified the graph in response to this request. At the request of another referrer, we have modified the graph into a histogram to make it clearer.
Comment 8 : Figure 1 – if units (%) are provided in the axis title, there is no need to include units on the chart. Additionally, I suggest starting OY with 4%, then the chart will be more readable. Similarly Figure 2.
Answer 8: We have modified the graph (1 and 2) in response to this request. At the request of another referrer, we have modified the graph (1 and 2) into a histogram to make it clearer.
Comment 8 : • Discussion is insufficient. First of all, the impact of socio-economic factors should be described in more detail.
Answer 8: We have modified our discussion to respond to this comment, in particular by discussing the socio-economic factors studied.
Comment 9 : • Delete point 6. Patents
Answer 9: We have delete point 6
Reviewer 2 Report
Comments and Suggestions for Authors
Minor editing of English language is required
Comments on the Quality of English Language
My specific comments on the manuscript are as follows:
In the introduction part - is brief, it would be appropriate to add some information about ISO-BMI, or methods for assessing obesity in children and adolescents... In the material and methods section: - it would be appropriate to add the number and name of the protocol of the ethics committee or the institution that approved the research.
Results - in the results, it is necessary to add other parameters for the assessment of overweight and obesity in children and adolescents, the assessment of BMI alone is not sufficient. Discussion - I have no comments Limits of the study - the mentioned limitations of the study are of a relatively serious nature, as in all cases they are important factors involved in the development of overweight and obesity. It would be appropriate to use it in the assessment of overweight and obesity in children and adolescents. In current form is the study methodologically oriented only to the assessment of BMI and the influence of parents' employment. I consider the above to be the greatest shortcoming of the submitted manuscript. Conclusion - I have no comments. References - it would be appropriate to add a few sources of discussion.
In the manuscript were used 24 literary sources (including 3 self-citations), of which 15 are from the last 5 years; 6 for the last 5-10 years and 3 older than 10 years.
Author Response
Thank you for your comments. We have tried to address the issue you have raised, as far as possible.
Comment 1: In the introduction part - is brief, it would be appropriate to add some information about ISO-BMI, or methods for assessing obesity in children and adolescents...
Answer 1 : We have added more information on the latest scientific developments on this subject in the introduction.
Comment 2: In the material and methods section: - it would be appropriate to add the number and name of the protocol of the ethics committee or the institution that approved the research.
Answer 2 : we have written a section (2.5) detailing how this study was accepted by the Geneva ethics committee. We added also the number.
Comment 3: Results - in the results, it is necessary to add other parameters for the assessment of overweight and obesity in children and adolescents, the assessment of BMI alone is not sufficient
Answer 3 : We are agree with this comment. We discuss other parameters for a more complete assessment of obesity in children in the introduction. Unfortunately, the only anthropometric measure available in this study was BMI. This is one of the limitations of our study, which we discuss below.
Comment 4 : Limits of the study - the mentioned limitations of the study are of a relatively serious nature, as in all cases they are important factors involved in the development of overweight and obesity. It would be appropriate to use it in the assessment of overweight and obesity in children and adolescents. In current form is the study methodologically oriented only to the assessment of BMI and the influence of parents' employment. I consider the above to be the greatest shortcoming of the submitted manuscript.
Answer 4 : We totally agree with this comment. Our study suffers from a major limitation in that we were only able to study BMI and its relationship with conventional socio-economic factors. Many other parameters would have been relevant to explore this problem in detail. We hope to be able to carry out another study on the subject in a more comprehensive and complex way.
Round 2
Reviewer 2 Report
Comments and Suggestions for Authors
I appreciate the efforts of the authors to consider my comments on the manuscript. As it is not possible to add other measurable indicators of obesity and the authors declare this in the limitation of the study, it is possible to publish the manuscript in its current form. It will affect the quality and informative value of the results achieved, but in my opinion it is publishable.
Author Response
Thank you for your comment. As I said earlier, we are aware of the limitations of this study, which only looked at BMI as an indicator of obesity, and not other anthropometric indicators. But this study will give us a reference point for a larger study that we want to carry out, which this time will tackle this issue in a more global way by including these other parameters relating to obesity.